# Generation of Non-Nucleotide CD73 Inhibitors Using a Molecular Docking and 3D-QSAR Approach

**DOI:** 10.3390/ijms222312745

**Published:** 2021-11-25

**Authors:** Swapnil P. Bhujbal, Jung-Mi Hah

**Affiliations:** 1College of Pharmacy, Hanyang University, Ansan 426-791, Korea; swapnil18@hanyang.ac.kr; 2Institute of Pharmaceutical Science and Technology, Hanyang University, Ansan 426-791, Korea

**Keywords:** chemotherapy, cancer, CD73, molecular docking, CoMFA, CoMSIA, 3D-QSAR, inhibitors

## Abstract

Radiotherapy and chemotherapy are conventional cancer treatments. Around 60% of all patients who are diagnosed with cancer receive radio- or chemotherapy in combination with surgery during their disease. Only a few patients respond to the blockage of immune checkpoints alone, or in combination therapy, because their tumours might not be immunogenic. Under these circumstances, an increasing level of extracellular adenosine via the activation of ecto-5’-nucleotidase (CD73) and consequent adenosine receptor signalling is a typical mechanism that tumours use to evade immune surveillance. CD73 is responsible for the conversion of adenosine monophosphate to adenosine. CD73 is overexpressed in various tumour types. Hence, targetting CD73’s signalling is important for the reversal of adenosine-facilitated immune suppression. In this study, we selected a potent series of the non-nucleotide small molecule inhibitors of CD73. Molecular docking studies were performed in order to examine the binding mode of the inhibitors inside the active site of CD73 and 3D-QSAR was used to study the structure–activity relationship. The obtained CoMFA (*q*^2^ = 0.844, ONC = 5, *r*^2^ = 0.947) and CoMSIA (*q*^2^ = 0.804, ONC = 4, *r*^2^ = 0.954) models showed reasonable statistical values. The 3D-QSAR contour map analysis revealed useful structural characteristics that were needed to modify non-nucleotide small molecule inhibitors. We used the structural information from the overall docking and 3D-QSAR results to design new, potent CD73 non-nucleotide inhibitors. The newly designed CD73 inhibitors exhibited higher activity (predicted pIC_50_) than the most active compound of all of the derivatives that were selected for this study. Further experimental studies are needed in order to validate the new CD73 inhibitors.

## 1. Introduction

Ecto-5’-nucleotidase (eN or e5NT) is a glycosylphosphatidylinositol (GPI)-anchored cell surface protein that is encoded by the NT5E gene [1,2,3]. It was named the cluster of differentiation 73 (CD73) in 1989. Human CD73 is usually expressed on lymphocytes, endothelial and epithelial cells [2]. The CD73 structure is composed of two identical 70 kDa polypeptide chains (a homodimer). Each monomer of CD73 is formed by two discrete domains: the N-terminal domain and the C-terminal domain, which are essential for the binding of two metal ions for catalysis and attaching the nucleotide substrate, respectively [2,3]. These two domains are connected by a single helix (residues 318–336) containing a small hinge region, enabling the enzyme to possess an open (butterfly-like) or closed conformation. The active site is situated at the interface between the two domains [2,4,5].

CD73 plays a crucial role in regulating adenosinergic signalling [2,3]. It is involved primarily in the formation of adenosine via the hydrolysis of extracellular adenosine-5’-monophosphate (AMP) but also in cell–cell and cell–matrix interactions, in addition to transmembrane signalling [6,7]. CD73 serves as an immune checkpoint by producing adenosine (ADO), which suppresses immune activation through the A2A and A2B receptors found on T cells, natural killer cells, and macrophages [8]. CD73 is overexpressed in several cancers, resulting in a higher level of ADO, which links to poorer patient prognosis [9,10,11].

Moreover, the increased concentration of ADO in the tumour microenvironment (TME) inactivates natural killer cells and reduces cytotoxic T-cell responses [12]. These events ultimately lead to impaired cancer immunosurveillance, encouraging tumour growth. The critical role of CD73 in ADO production indicates that CD73/adenosine signalling is an attractive target for the reversal of adenosine-mediated immune suppression [1,10,13]. The overexpression of CD73 has been reported to cause different types of cancers such as breast cancer, colorectal cancer, NSLC lung cancer, papillary thyroid cancer, and melanoma [10]. Therefore, the design of new potent inhibitors is an important endeavour in the process of CD73 drug discovery.

Substantial development of the design of nucleotide and non-nucleotide inhibitors of CD73 has been reported. A derivative of ADP, AMPCP (Ki of 0.87 μM), was the first promising CD73 inhibitor [14,15]. In the early stages of CD73 drug discovery, modifying the AMPCP scaffold was the main strategy applied by many pharmaceutical companies and several research groups, which has led to the development of compounds such as PSB-12379 (Ki of 2.21 nM), PSB-1489 (Ki of 0.32 nM), PSB-12646 (Ki of 15.5 nM), PSB-12604 3 (Ki of 184 nM), AB680 (Ki of 0.005 nM), and OP-5244 (Ki of 0.25 nM) [15,16]. Among these, AB680 is in the early phase of clinical trials as an anticancer immunotherapeutic agent [9,16].

Similarly, Müller et al. have proposed non-nucleotide inhibitors that are based on the availability of the X-ray crystal structure of human CD73. Baicalin is a natural product that first was co-crystallised with human CD73 as its inhibitor [6]. Studies by Müller et al. and Bajorath et al. used virtual screening and identified non-nucleotide sulphonamide-based CD73 inhibitors, of which 6-chloro-2-oxo-N-(4-sulfamoylphenyl)-2H-chromene-3-carboxylic acid amide (Ki = 287 nM) was a potent compound [4,17]. Anthraquinone sulfonates are another interesting group of inhibitors reported by Müller et al. and they include PSB-0952 (Ki of 260 nM) and PSB-0963 (Ki of 150 nM) [2]. Of all of the known non-nucleotide compounds, 4-({5-[4-fluoro-1-(2 H-indazol-6-yl)-1H-1,2,3-benzotriazol-6-yl]-1H-pyrazol-1-yl}methyl)benzonitrile was found to have the best inhibitory activity (IC_50_ of 12 nM) [2].

All of these compounds were reported to be moderately to highly potent and competitive inhibitors of CD73, but the challenge to discover less acidic, non-nucleotide small molecule inhibitors has been restricted to a certain group of structural families, including sulphonamides, anthraquinones, and other flavonoids [10]. Hence, the development of alternative chemotypes for small molecule inhibitors of CD73 holds great promise. Nevertheless, there are a few monoclonal antibodies, such as MEDI9447 (AstraZeneca), BMS-986179 (Bristol-Meyer Squibb), SRF373/NZV930, CPI-006/CPX-006, IPH5301, and TJ004309, which have entered Phase I/II clinical trials possessing antitumor activity against CD73 for the treatment of tumours in areas of the body such as the breast, pancreas, prostate, and lung [1,3]. However, small molecule inhibitors have the advantage of use in oral administration. Therefore, we focused on designing more potent, small molecule non-nucleotide CD73 inhibitors.

Structure–activity relationship studies of non-phosphonate CD73 inhibitors using the three-dimensional quantitative structure–activity relationship (3D-QSAR) have not been reported to date. They can reveal detailed structural information by using the comparative molecular field analysis (CoMFA) and comparative molecular similarity indices analysis (CoMSIA) descriptor fields (such as steric, electrostatic, hydrophobic, and hydrogen bond donor and acceptor), this structural information is helpful in the design of new inhibitors. Hence, in this study, we selected a novel series of small molecule non-nucleotide derivatives as CD73 inhibitors, which are reported to be more potent than all other non-phosphonate inhibitor classes to date. We performed molecular docking and 3D-QSAR studies on these derivatives in order to study the binding mode and structure–activity relationship, which revealed important structural characteristics to design more potent non-nucleotide CD73 inhibitors. Overall, the docking analysis and structural insights from the 3D-QSAR contour maps directed us to design new non-nucleotide CD73 inhibitors that showed better inhibitory (pIC_50_) activity than the most active compound of the selected derivatives.

## 2. Results and Discussion

### 2.1. Molecular Docking

Molecular docking was used in order to understand the binding mode of the most active compound, **48**, inside the active site of CD73. We used a standard protocol to perform induced fit docking, which generated 20 binding poses for compound **48**. All 20 poses were analysed for their docking score and their bonding and non-bonding interactions. One of the poses showed a docking score of −10.43 and possessed a binding pose with CD73 that was similar to that observed between the co-crystallised ligand and the protein (PDB ID: 6XUE) (Figure 1). Hence, this pose was selected and further interactions were examined. Compound **48** was the most active and docked inside the binding pocket to form three hydrogen bond (H-bond) interactions with CD73. The indazole ring of compound **48** was docked towards the dimetallic catalytic centre that formed an H-bond interaction between the H atom of the NH group of indazole and ASP506. This interaction was reported to be one of the important interactions for the inhibition of CD73, because ASP506 lies near the catalytic site of CD73 [1]. Another H-bond interaction was observed between the N2 atom of the benzotriazole core and active site residue ASN390. Similarly, the N2 atom of the pyrazole moiety formed an H-bond interaction with ASN186. Additionally, the benzotriazole core of compound **48** was aligned, edge to face, with the residues PHE500 and PHE417 and they formed pi–pi stacks. These two residues serve as an ‘adenine clamp’ when binding to the native substrate, AMP, and bind to the nucleotide CD73 inhibitor AMPCP. Thus, these pi–pi interactions are crucial in the binding mechanism. There was also a pi–cation interaction observed between the indazole ring and ARG354. Overall, the interactions were similar to those observed between a co-crystallised moderately active compound of this dataset (compound **49**) and CD73 [1]. Therefore, the docked pose of compound **48** substantiates a strong binding conformation.

The docked pose was analysed in order to assess the hydrophobic interactions. A python script called ‘colour h’ was used to colour the hydrophobic residues of CD73 in order to assess their interactions with compound **48**. This script uses an Eisenberg hydrophobicity scale (Figure 2) to colour the receptor in PyMOL [18]. It provides colouring from red (for the most hydrophobic residues) to white (for the least hydrophobic region). The indazole part of the ligand was docked inside the hydrophobic pocket, which formed hydrophobic interactions with residues GLY392, GLY393, and GLY447. The benzotriazole core of compound **48** appeared to form interactions with PHE417. Moreover, the benzonitrile part of the compound was in close proximity to the hydrophobic residues PRO182, PHE183, LEU184, PHE500, and GLY505, forming hydrophobic interactions. Overall, docking analysis suggests that the selected docked conformation of the most active compound (**48**) was adequate and it was therefore used in further 3D-QSAR studies.

### 2.2. 3D-QSAR

Receptor-based 3D-QSAR, CoMFA, and CoMSIA models were developed for small molecule non-nucleotide CD73 derivatives. The dataset compounds were sketched and aligned inside the binding site of the receptor using the docked conformation of compound **48** as a template in SYBYL-X, version 2.1. The alignment of the dataset compounds is shown in Figure 3. The dataset was divided into training (42) and test (14) sets, using the standards proposed in Algorithm 4 (activity ranking) in an earlier article [19]. Consequently, our test set comprises compounds with high, moderate, and low activity (pIC_50_) values. The test set compounds are indicated by * in Appendix A.

The reliability of a 3D-QSAR model is assessed by calculating various statistical parameters such as the cross-validated correlation coefficient (*q*^2^), non-cross validated correlation coefficient (*r*^2^), standard error of estimate (SEE), optimal number of components (ONC), and F value by applying partial least square (PLS). We generated the CoMFA models (*q*^2^ = 0.763, ONC = 5, and *r*^2^ = 0.915) for all dataset compounds (training + test set) and (*q*^2^ = 0.844, ONC = 5, and *r*^2^ = 0.947) for the chosen test set 5. The model with the best *q*^2^ and *r*^2^ values was selected as the final model. Likewise, the CoMSIA models were established with different field combinations and are depicted in Appendix A. A combination of steric, hydrogen bond acceptor and donor, and hydrophobic (SHAD) fields yielded a CoMSIA model with satisfactory statistical values (*q*^2^ = 0.760, ONC = 6, and *r*^2^ = 0.943). However, the CoMSIA model that was obtained using an external test set showed better results (*q*^2^ = 0.804, ONC = 4, and *r*^2^ = 0.954) and was used for further validation. The detailed statistical values of the selected CoMFA and CoMSIA models are illustrated in Table 1. In Table 1, the models under the ‘Full Model’ section represent the models that were used for the whole dataset (training + test sets) and the models under the ‘Test Set 5’ section represent the models that were obtained using test set 5.

#### Validation of 3D-QSAR Models

The derived CoMFA and CoMSIA models were validated using a variety of validation techniques in order to assess their predictive ability and robustness. All techniques, such as predictive *r*^2^ (*r*^2^*_pred_*), bootstrapping, progressive scrambling (*Q*^2^), and leave-out-five (LOF), showed statistical values that were within the acceptable range [20,21]. These values verified that the models were robust and predictive and are shown in Table 1. The residual values, along with the experimental and predicted activity values, for the selected models are shown in Appendix A. The scatter plots for these are illustrated in Figure 4.

### 2.3. Contour Map Analysis

#### 2.3.1. CoMFA Contour Maps

The steric and electrostatic contour maps of the CoMFA model are shown superimposed with that of compound **48** in Figure 5A,B, respectively. The green and blue-coloured contour maps depict the favourable regions for steric and electropositive substitutions, respectively; whereas yellow and red denote unfavourable regions.

Two green contours (Figure 5A) were noted around the benzonitrile ring at the R^1^ position, suggesting that the bulky groups are favoured at this region in order to improve potency. A steric group at the R^1^ position could interact with some hydrophobic residues of CD73. This can be validated by hydrophobic interactions with residues PRO182, PHE183, and GLY505, as observed in our docking analysis of the most active compound, **48**. There were also yellow contours around the benzonitrile ring, suggesting that this region is unfavourable for the bulky groups, although this seems to be position-specific. Similarly, one small yellow contour was observed near the indazole ring, indicating an unfavourable region for bulky substitution.

In the electrostatic contour map (Figure 5B), two large, blue-coloured contours were seen around the benzotriazole core near the R^2^ position, which indicates that this is a favourable position for the electropositive group. H-bond interaction can occur with the residue ASN390 that can be verified by the occurrence of an H-bond interaction between the N2 atom of the benzotriazole core and ASN390 from docking analysis. This could be the reason for the moderate to high activities of compounds **42**, **49**, and **50** and the most active compound, **48**. On the other hand, one red contour was observed near the benzonitrile ring at the R^1^ position, signifying that this is a favourable place for electronegative groups.

#### 2.3.2. CoMSIA Contour Maps

The combination of the SHAD fields was used to develop the CoMSIA contour maps, which are shown in Figure 6. We discuss only the hydrophobic hydrogen bond donor and acceptor contour maps below, since the CoMSIA steric contour is similar to that of CoMFA. Figure 6B shows the hydrophobic contour map, in which magenta contours denote favourable regions for hydrophobic substitution, whereas orange contours denote unfavourable regions. One small magenta contour is present near the benzonitrile ring, which illustrates that presence of hydrophobic groups at this position, which could elevate the activity of the compound. This can be verified by the hydrophobic interactions of this ring with the residues PRO182, PHE183, and GLY505 that were observed in our docking analysis of the most active compound, **48**. Additionally, it could be the reason why compounds **28**, **42**, **47**, **49**, **50**, and **54** possess moderate to high activity. The two maps with orange contours that were found near the benzotriazole core and indazole ring explain that substituting hydrophobic groups at these locations can decrease the potency of a compound.

In the hydrogen bond acceptor contour (Figure 6C), the favourable and unfavourable regions are denoted by blue and red contour maps, respectively. Two small blue-coloured contour maps were present near the benzonitrile ring and benzotriazole core. The presence of the blue contour near the benzotriazole core suggests that the H-bond acceptor group is favourable at this place. This was supported by the occurrence of an H-bond interaction between the N2 atom of the benzotriazole core and ASN390. Red-coloured contours were noted near the indazole ring and pyrazole ring, signifying that the H-bond acceptor groups are not favourable at this position.

The hydrogen bond donor contour map is shown in Figure 6D. Here, cyan contours signify the regions where H-bond donor groups are favourable in enhancing inhibitory activity, whereas purple contours signify H-bond donor-unfavourable regions. The cyan contours that are present near the indazole ring suggest that this position is favourable for H-bond donor groups to increase the potency of a compound. The H-bond between the NH of the indazole ring and residue ASP506 from our docking analysis supports the presence of a cyan-coloured contour at this place. This can be observed by the moderate to high activities of compounds **8**, **22**, **42**, **49**, and **54** and (the most active compound) **48**, which possesses H-bond donor groups at this position. The purple-coloured contours that are near the benzotriazole core and pyrazole ring do not encourage the addition of H-bond donor groups at this position.

### 2.4. Designing New CD73 Inhibitors

The developed 3D-QSAR models, CoMFA, and CoMSIA (SHAD) revealed important structural characteristics in terms of steric, electrostatic, hydrophobic, and H-bond donor and acceptor fields. This structural information and the crucial interactions observed in the docking analysis of the most active compound (**48**) were used to derive a drug design strategy to design new non-nucleotide CD73 inhibitors (Figure 7). According to this design strategy, we modified different substituents at position R^1^ while keeping other substitutions unchanged in order to assess the difference in the activity. We followed the same procedure for each position in order to determine which substituent is more potent at that particular position. We found that 4-trimethylpyrido [3,2-d]pyrimidine and pyrano [3,4-c]pyrazole at the R^1^ position; 2-methyl-1,3-thiazinane and phenol at the R^2^ position; 1-chloro-2-((trifluoromethyl)sulfonyl)benzene, 1-chloro-3-((trifluoromethyl)sulfonyl)benzene, 1-bromo-2-((trifluoromethyl)sulfonyl)benzene and 1-bromo-3-((trifluoromethyl)sulfonyl)benzene at the R^3^ position; and isoxazole, 1H-1,2,3-triazole, 1,2,5-thiadiazole, and pyridine at the R^4^ position possessed better activity (pIC_50_) than the most active compound. Finally, we chose these individual substituents from each position (which showed equal or better predicted activity than the most active compound) and tried different combinations in order to obtain a new scaffold as a more potent CD73 inhibitor. The structures and the predicted pIC_50_ values of the newly designed compounds are presented in Appendix A and a few selected compounds are presented in Table 2 below. The SMILES codes for the same are given in Appendix A (CD73 Designed Compounds) in the Appendix A.

Furthermore, we have calculated the ADMET properties for all of the designed compounds. The detailed properties for all of the compounds are shown in Appendix A. We have calculated in silico ADMET (absorption, distribution, metabolism, excretion, and toxicity), physicochemical properties, pharmacokinetics, drug-likeness and medicinal chemistry friendliness using the SwissADME [22] web tool and pkCSM [23] (Appendix A). For lipophilicity, XLOGP3 should be in the range from −0.7 to +6.0. For solubility, log S (calculated with the ESOL model36) should not exceed 6. A qualitative estimation of the solubility class is given according to the following log S scale: insoluble < −10 < poorly < −6 < moderately < −4 < soluble < −2 < very < 0 < highly. The Synthetic Accessibility (SA) score ranges from 1 (very easy) to 10 (very difficult). Toxicity predicts whether a given drug is likely to be Ames-positive and, hence, mutagenic. Drug-likeness evaluates, qualitatively, the chance for a molecule to become an oral drug with respect to bioavailability. Violation of Lipinski’s rule-of-five filter defines four classes of compounds with probabilities of 11%, 17%, 56% and 85%. Hence, the prediction results that are depicted in Appendix A show that the designed inhibitors possess promising ADMET properties. The molecular docking of all the designed compounds was performed as well, using the same method that is used to dock the most active compound of the dataset, using Schrödinger Maestro, version 12.8. Almost all of the designed compounds showed similar binding poses and promising docking scores that were similar to those of the most active compound from the dataset. The detailed docking analysis is shown in Appendix A and the representative docked pose of the compound D13 is shown in Appendix A in the Appendix A.

## 3. Materials and Methods

### 3.1. Training Set/Test Set Selection for 3D-QSAR Analyses

The design and synthesis of CD73 non-nucleotide inhibitors are not often explored using the 3D-QSAR technique. Thus, we selected a series of non-nucleotide small molecule inhibitors of CD73 that mimic the ionizable and acidic structure of AMP, the usual substrate of the enzyme [1]. The selected dataset comprises 56 non-nucleotide derivatives spanning the log value of more than 4 logarithmic units, which is within the prerequisite range [24]. All of the structures of the compounds were drawn using the sketch module in SYBYL-X version 2.1 and were optimised using energy minimization with the Tripos force field [25]. The biological activities (IC_50_) were modified into pIC_50_ (-log IC_50_) values. The pIC_50_ values were used as dependent variables in order to derive the 3D-QSAR models. The dataset was divided into a training set of 42 compounds for model generation and 14 compounds as a test set for model validation, based on the structure and activity span of the compounds. The test set compounds were chosen to exhibit compounds with low, medium, and high activity as suggested in Algorithm 4 [19]. Compounds with undefined activity values were eliminated as outliers. The chemical structures of the dataset compounded with their IC_50_ values are shown in Appendix A. The SMILES codes for the same are given in Appendix A (CD73 Dataset Compounds) in the Appendix A.

### 3.2. Molecular Docking

Schrödinger Maestro, version12.8 [26] (Release 2021-2, Schrödinger, LLC, New York, NY, USA), was used to perform the molecular docking of the most active compound (**48**) of the selected dataset [27]. The structure of compound **48** was drawn using Chemdraw [28] and its 3D conformation was generated using the Schrödinger LigPrep programme [29]. LigPrep generated all possible tautomers and states at pH 7.0 using Epik [30] for compound **48** and the specified chiralities were retained following minimization using the OPLS 2005 force field [27]. The crystal structure of CD73 co-crystallised with one of the dataset compounds **49** (PDB ID: 6XUE) was acquired from the Protein Data Bank (PDB) [1]. The protein was prepared using the Protein Preparation Wizard [31] to assign bond orders, add hydrogens at pH 7.0, and remove water molecules. Prime was used to complete missing side chains and loops. Finally, a restrained minimization was performed using the default constraint of 0.30 Å RMSD and the OPLS 2005 force field in order to complete the protein preparation. Molecular docking simulations were performed using the Glide induced fit docking module [32] in standard protocol (standard precision) mode. The binding conformations of compound **48** were analysed in order to identify the important interactions with the active site residues of CD73. The selected docked pose of compound **48** was used as a template in 3D-QSAR model generations.

### 3.3. Receptor-Based CoMFA and CoMSIA Models

We obtained the 3D-QSAR models by using CoMFA [33] and CoMSIA [34] to correlate the 3D structures of non-nucleotide inhibitors and biological activity using SYBYL-X, version 2.1 [34,35]. The alignment of the dataset compounds was carried out inside the active site of the receptor using a common scaffold alignment method, taking the most active compound, **48**, as a template molecule. CoMFA uses the steric and electrostatic potential energies that are computed with the help of the Lennard–Jones and Coulombic potentials, respectively [33].

The application of a suitable partial charge is important in order to attain reasonable 3D-QSAR models. We used Pullman as a partial charge scheme [36] and the default parameters to develop our 3D-QSAR models. A grid spacing of 2.0 Å and an sp^3^ hybridised carbon as a probe atom with +1 charge were used. PLS regression was used in order to obtain statistically reasonable CoMFA models. The CoMFA descriptors were used as independent variables and the biological activity values (pIC_50_) were used as dependent variables in PLS analyses. PLS analysis with leave-one-out cross-validation was performed to assess the reliability of the generated models and to calculate the squared cross-validated correlation coefficient (*q*^2^) value, an ONC, and the standard deviation of prediction (SEP). A column filtering value of 2.0 and the obtained ONC were used in non-cross-validation analysis to calculate the squared correlation coefficient (*r*^2^), F-test value (F) and standard error of estimation (SEE).

However, CoMSIA uses descriptors such as steric, electrostatic, hydrophobic, and hydrogen bond donor and acceptor. These five CoMSIA similarity indices were calculated by using a probe atom of radius 1.0 Å and an attenuation factor of 0.30. A Gaussian function was used to compute the CoMSIA model between the grid point and each atom of the molecule. Several CoMSIA models were derived from the diverse descriptor combinations using the same lattice box that was used in CoMFA. The model that showed reasonable *q*^2^ and *r*^2^ statistical values was selected as the final model. The selected models were further validated using various validation methods.

#### 3D-QSAR Model Validation

Numerous validation techniques such as bootstrapping, progressive scrambling, LOF (leave-out-five), and external test set validation were executed in order to assess the stability, robustness, and predictive ability of the derived models. To evaluate the model’s reliability, bootstrapping for 100 runs and the progressive scrambling of 10 samplings with 2–10 bins were performed [37]. The predictive ability that is expressed by the predictive correlation coefficient (*r*^2^*_pred_*) was calculated using the formula given below:*r*^2^*_pred_* = (SD − PRESS)/SD(1)
where SD is the sum of the squared deviations of each experimental value from the mean and PRESS is the sum of the squared differences between the predicted and actual affinity values.

The standard contour maps were obtained for both the CoMFA and CoMSIA models. The structural information that was revealed from the analysis of the contour maps was applied in order to derive a new design strategy and we designed more potent CD73 non-nucleotide inhibitors.

### 3.4. Design of New CD73 Inhibitors and ADMET Calculation

We have derived a design strategy based on the structural information that was obtained from the contour map analyses of selected CoMFA and CoMSIA (SHAD) models. We designed 52 new compounds and further calculated their in silico ADMET (absorption, distribution, metabolism, excretion, and toxicity) pharmacokinetic properties using online tools such as SwissADME and pkCSM. SwissADME is a web tool that gives free access to a pool of fast, yet robust, predictive models that can be used for assessing the physicochemical properties, pharmacokinetics, drug-likeness and medicinal chemistry friendliness of a ligand (drug molecule). It provides a user-friendly interface through the login-free website http://www.swissadme.ch. accessed on 5 Novermber 2021 The SMILES code for each compound was given as an input in the SwissADME tool. Consequently, the calculated properties were displayed on the output webpage. Similarly, pkCSM was used to calculate the toxicity of the designed compounds. It is a freely accessible web server (http://structure.bioc.cam.ac.uk/pkcsm accessed on 5 Novermber 2021) and provides an integrated platform to rapidly evaluate pharmacokinetic and toxicity properties. It uses graph-based signatures in order to develop predictive models of central ADMET properties for drug development.

## 4. Conclusions

CD73 is one of the ectonucleotidases that plays a crucial role in regulating adenosinergic signalling and the conversion of extracellular AMP to adenosine. The overexpression of CD73 is associated with various tumour types, such as breast cancer, colorectal cancer, prostate cancer, pancreatic cancer, etc. Due to its crucial role in the regulation of key signalling molecules, CD73 is a significant and competitive target for the design of new, potent anticancer inhibitors. In our study, we used different modelling techniques, such as molecular docking in order to investigate and find the crucial active site residues that are responsible for the inhibition of CD73 and 3D-QSAR in order to study the structure-activity relationship. The docking analysis revealed crucial active site residues for CD73 inhibition, such as ASN186, ARG354, ASN390, PHE417, PHE500 and ASP506. Moreover, CoMFA (q^2^ = 0.844, ONC = 5, and r^2^ = 0.947) and CoMSIA (SHAD) (q^2^ = 0.804, ONC = 4, and r^2^ = 0.954) produced satisfactory statistical models in terms of q^2^ and r^2^. The validation of these models verified that they are predictive and reliable. The structural information that was obtained from the analysis of the contour maps of 3D-QSAR models was consistent with the overall interactions observed in the docking analysis. These results elucidated the detailed structural properties, providing information that may assist in modifying the most active compound of the selected non-nucleotide derivatives and designing new, potent CD73 inhibitors. Thus, we developed a design strategy in order to obtain more CD73 antagonists with a new scaffold, possessing better predicted activity (pIC_50_) values than the most active compound of the dataset. We designed and predicted the activity of around 52 new CD73 non-nucleotide inhibitors, which were more active than the selected dataset compounds in this study. The overall results of our study reveal insights for medicinal chemists that they may use to synthesise more potent non-nucleotide small molecule CD73 inhibitors. Further experimental studies need to be performed on the designed compounds in order to check their pharmacodynamics/pharmacokinetic properties.

## Figures and Tables

**Figure 1 ijms-22-12745-f001:**
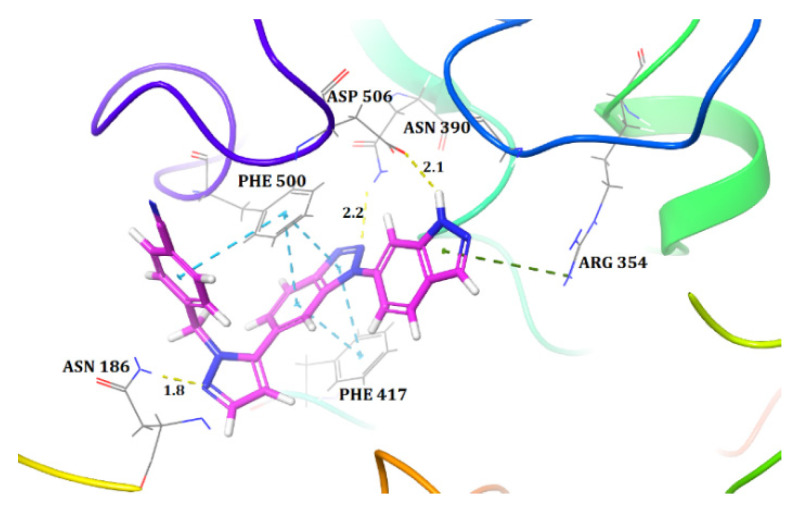
Docked pose of the most active compound, **48**, inside the active site of CD73 (Hydrogen bonds are represented as yellow dotted lines, pi–pi interactions are represented as cyan dotted lines, and pi–cation interactions are represented as green dotted lines).

**Figure 2 ijms-22-12745-f002:**
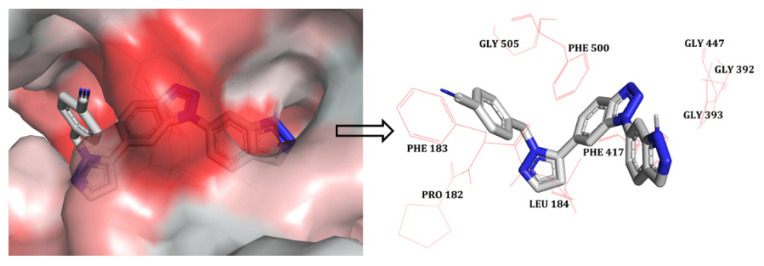
The most active compound, **48**, (shown in stick model) inside the hydrophobic pocket of CD73; The red-coloured region represents the most hydrophobic surface of the protein, and the white colour represents the least hydrophobic surface. Hydrophobic residues are indicated with red lines.

**Figure 3 ijms-22-12745-f003:**
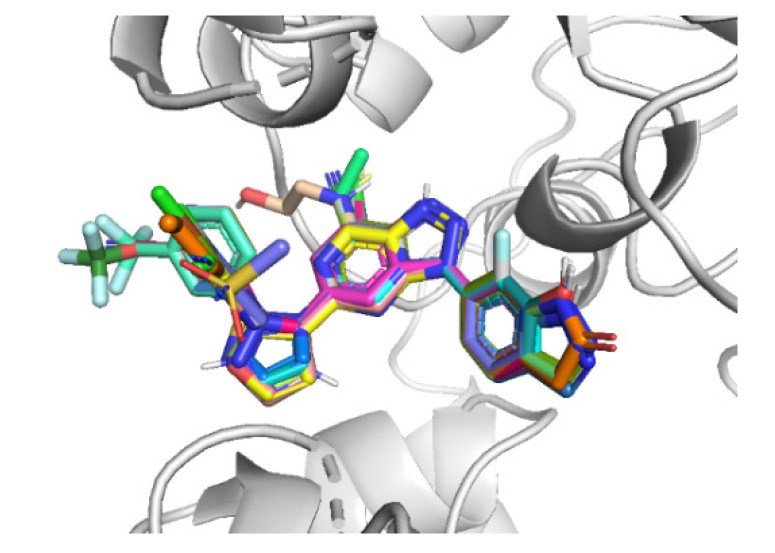
Alignment of the dataset compounds inside the active site of CD73.

**Figure 4 ijms-22-12745-f004:**
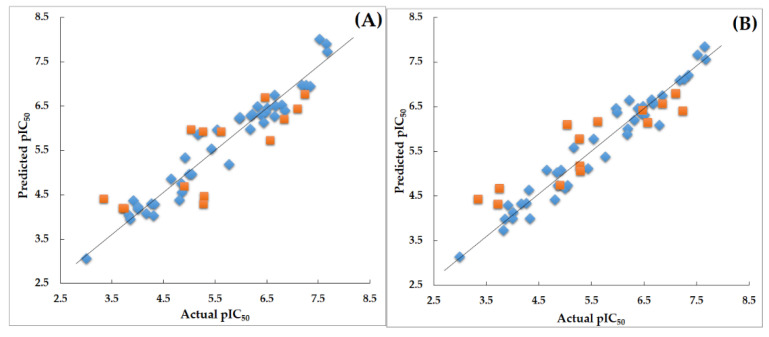
(**A**) Scatter plot for the selected CoMFA model; (**B**) Scatter plot for the selected CoMSIA model; the plot shows the actual pIC_50_ versus predicted pIC_50_ activity of training and test sets; the training set compounds are represented as blue diamonds; the test set compounds are represented as orange squares.

**Figure 5 ijms-22-12745-f005:**
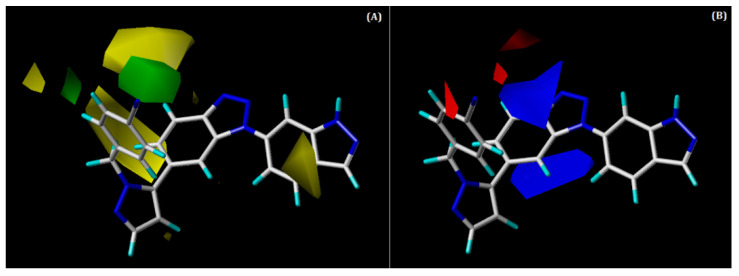
Contour maps for the selected CoMFA model. (**A**) Steric contour map; (**B**) electrostatic contour map; green contour shows the region is favourable for bulky substitutions, and yellow contours show the region is unfavourable; blue contours favour electropositive substitutions, whereas red contours favour electronegative substitutions.

**Figure 6 ijms-22-12745-f006:**
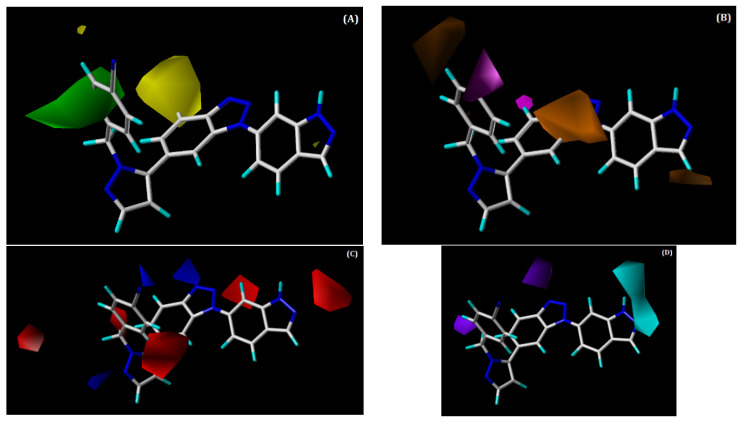
Contour maps for the selected CoMSIA model. (**A**) Steric contour map; (**B**) hydrophobic contour map; (**C**) hydrogen bond acceptor contour map; (**D**) hydrogen bond donor contour map. Green contour shows regions that are favourable for bulky substitutions, whereas yellow contours show unfavourable regions; magenta contour shows the region favourable for hydrophobic substitutions, whereas orange contours show unfavourable regions; blue contours favour H-bond acceptor, while red contour does not favour H-bond acceptor substitutions. The favourable region of the hydrogen bond donor contour map is depicted by cyan, whereas purple denotes the opposite.

**Figure 7 ijms-22-12745-f007:**
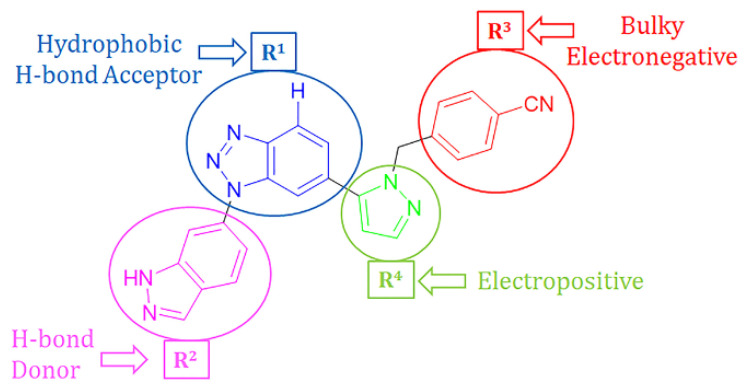
The design strategy for CD73 non-nucleotide small inhibitor design.

**Table 1 ijms-22-12745-t001:** Detailed statistical values of the selected CoMFA and CoMSIA models.

Parameter	Full Model		Test Set 5	
	CoMFA	CoMSIA (SHAD)	CoMFA	CoMSIA (SHAD)
*q* ^2^	0.763	0.760	0.844	0.804
ONC	5	6	5	4
SEP	0.635	0.646	0.525	0.581
*r* ^2^	0.915	0.943	0.947	0.954
SEE	0.380	0.314	0.306	0.282
F value	108.134	134.938	129.146	191.403
LOF	-	-	0.848	0.771
BS-*r*^2^	-	-	0.968	0.961
BS-SD	-	-	0.012	0.012
*Q* ^2^	-	-	0.626	0.599
*r* ^2^ * _pred_ *	-	-	0.698	0.757

*q*^2^: squared cross-validated correlation coefficient; ONC: optimal number of components; SEP: standard error of prediction; *r*^2^: squared correlation coefficient; SEE: standard error of estimation; F value: F-test value; LOF: leave-out-five; BS-*r*^2^: bootstrapping *r*^2^ mean; BS-SD: bootstrapping standard deviation; *Q*^2^: progressive sampling; *r*^2^*_pred_*: predictive *r*^2^.

**Table 2 ijms-22-12745-t002:** The structures and the predicted pIC_50_ values of selected new designed CD73 inhibitors.

Compound	R^1^	R^2^	R^3^	R^4^	Predicted pIC_50_
D5	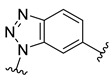	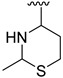	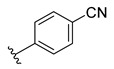	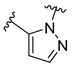	7.603
D6	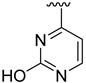	7.540
D8	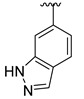	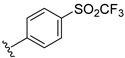	7.645
D11	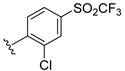	8.056
D12	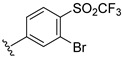	8.134
D15	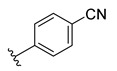	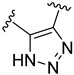	7.769
D20	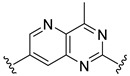	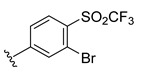	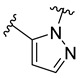	7.839
D24	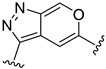	8.088
D33	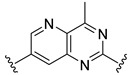	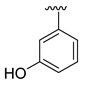	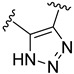	8.047
D45	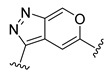	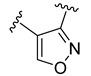	8.236
D50	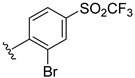	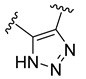	8.112

## Data Availability

Not applicable.

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
