# Peer review of "Generation of Non-Nucleotide CD73 Inhibitors Using a Molecular Docking and 3D-QSAR Approach"

_ijms, 2021, doi:10.3390/ijms222312745_

Round 1
Reviewer 1 Report
As mentioned by the authors, I was not aware of the scope of this special issue "Chemoinformatics and Bioinformatics Tools in Structure-Activity modelling in molecular sciences." If experimental validation of statistical data is not needed in this Special Issue, I recommend accepting the paper in its present form.
Author Response
As mentioned by the authors, I was not aware of the scope of this special issue "Chemoinformatics and Bioinformatics Tools in Structure-Activity modelling in molecular sciences." If experimental validation of statistical data is not needed in this Special Issue, I recommend accepting the paper in its present form. --> Appreciate it.
Reviewer 2 Report
Bhujbal and Hah describe describe the application of molecular docking and 3D-QSAR for analyses and design of novel CD73 inhibitors.
As I wrote in my previous review, the in silico analyses are well performed and the designed structures are promising. However, experimental confirmation of the activities are absent. As the topic of the Special issue Chemoinformatics and Bioinformatics Tools in Structure-Activity Modelling in Molecular Sciences is focused on the in silico methods and models for design and analysis of novel molecules, the ms seems suitable for publication in this issue.
I recommend the ms to be directed to the Special issue Editor Dr. Bono Lučić for final decision.
Author Response
Bhujbal and Hah describe describe the application of molecular docking and 3D-QSAR for analyses and design of novel CD73 inhibitors.
As I wrote in my previous review, the in silico analyses are well performed and the designed structures are promising. However, experimental confirmation of the activities are absent. As the topic of the Special issue Chemoinformatics and Bioinformatics Tools in Structure-Activity Modelling in Molecular Sciences is focused on the in silico methods and models for design and analysis of novel molecules, the ms seems suitable for publication in this issue. I recommend the ms to be directed to the Special issue Editor Dr. Bono Lučić for final decision. --> Appreciate it, and we respond to the Editor Dr. Bono Lučić 's comments in the revised manuscript.
Reviewer 3 Report
If experimental validation of statistical or computational data/results is not necessary for this Special Issue, I recommend accepting the manuscript in its present form.
Author Response
If experimental validation of statistical or computational data/results is not necessary for this Special Issue, I recommend accepting the manuscript in its present form. --> Appreciate it.
This manuscript is a resubmission of an earlier submission. The following is a list of the peer review reports and author responses from that submission.
Round 1
Reviewer 1 Report
The authors provide us with 52 novel non-nucleotide CD73 inhibitors. However, as mentioned by the authors, no experimental studies are performed on the designed compounds to validate their selective inhibitory capacity on CD73. These are major missing experiments.
Are these non-nucleotide inhibitors selective? Or do these compounds show any off target activity against the ecto-nucleotidase NTPDase-1 (CD39)?
What about the chemical stability of these compounds under both acidic and basic conditions?
As mentioned by the authors, CD73 is expressed by cancer cells and immune cells. What is the potency of these compounds against CD73 in these cells?
Reviewer 2 Report
Bhujbal and Hah describe the application of molecular docking and 3D-QSAR for analyses and design of novel CD73 inhibitors.
The in silico analyses are well performed and the designed structures are promising. However, experimental confirmation of the activities of some of the best predicted compounds are needed for publication in IJMS.
Reviewer 3 Report
Swapnil P. Bhujbal and Jung-Mi Hah in manuscript entitled “Generation of non-nucleotide CD73 inhibitors using a molecular docking and 3D-QSAR approach” studied newly designed CD73 inhibitors targetting CD73 signalling that can be use to block a typical mechanism that tumours use to evade immune surveillance. The study is important and interesting, but I do have one major comment. The authors present IC50 as constant value when in fact only Ki is constant that can be compared between different laboratories. IC50 depends from the lab routines and different conditions and as such cannot be compared with the results obtained by others. In the Introduction they show some inhibitors with their Kis and they should also show Kis of tested in MS compounds to have correct idea about their strength.
I have also a few minor comments:
- The abstract should be concentrated more on the subject of the study – methods, results and conclusion but now it looks more like introduction.
- Page 2, line 3 please change “through A2A and A2B receptors” to “through A2A and A2B receptors”.
- Page 16 the repetition of the figure 7 at the first raw of table 3 is not necessary and should be removed.
- In introduction and abstract there is a lot information about cancer but in fact study does not test any direct aspect of cancer – no assays with cancer cells or macrophages that may be influence by N5’N inhibitors etc. so it should be reduce to just information or such assays should be added to MS.
Reviewer 4 Report
The main issue with this manuscript is the lack of any experimental validation of the computation results. As such, the manuscript does not provide any new knowledge as it supplies only unconfirmed predictions. From this reason the manuscript should be rejected. Moreover, the selectivity of the compounds against potential off-targets should be experimentally tested.
In the present version, the manuscript containing only molecular modeling results, even technically sound, could be submitted to a more specialized journal focused on computer-aided drug design. It should be stressed, however, that the modelling methods used are classic and well-established so even such a journal may not find this manuscript worth publishing.